# Family-Level Impact of Germline Genetic Testing in Childhood Cancer: A Multi Family Member Interview Analysis

**DOI:** 10.3390/cancers17030517

**Published:** 2025-02-04

**Authors:** Sophie Van Hoyweghen, Kathleen B. M. Claes, Robin de Putter, Claire E. Wakefield, Marie Van Poucke, Marieke Van Schoors, Sabine Hellemans, Lesley Verhofstadt

**Affiliations:** 1Department of Biomolecular Medicine, Faculty of Medicine and Health Sciences, Ghent University, 9000 Ghent, Belgium; kathleen.claes@ugent.be (K.B.M.C.); robin.deputter@ugent.be (R.d.P.); 2Department of Experimental-Clinical and Health Psychology, Faculty of Psychology and Educational Sciences, Ghent University, 9000 Ghent, Belgium; lesley.verhofstadt@ugent.be; 3Cancer Research Institute Ghent CRIG, 9000 Ghent, Belgium; 4Center of Medical Genetics, Ghent University Hospital, 9000 Ghent, Belgium; marie.vanpoucke@uzgent.be; 5School of Clinical Medicine, UNSW Medicine and Health, UNSW Sydney, Sydney 2170, Australia; c.wakefield@unsw.edu.au; 6Behavioural Sciences Unit, Kids Cancer Centre, Sydney Children’s Hospital, Randwick 2031, Australia; 7Familieplatform VZW, 2600 Berchem, Belgium; marieke.vanschoors@familieplatform.be

**Keywords:** cancer, cancer predisposition, children, germline genetic testing, oncology, parents, pediatric oncology, psycho-oncology, family impact, multi-family member interview

## Abstract

Genetic testing is increasingly used in children with cancer to detect inherited genetic changes that may increase their risk for cancer. Many families are interested in this testing, but little is known about how it affects the entire family. This study explores how parents experience the impact of genetic testing for cancer risk on their family as a whole. We interviewed both parents in six families with a child diagnosed with cancer to understand their perspectives. The testing was performed by analyzing genes linked to cancer risk. Parents found genetic testing valuable and relatively straightforward but noted that it was hard to separate its effects from the emotional stress of their child’s cancer diagnosis and treatment. Despite this, they described how the testing influenced family life in significant ways. Key themes included how families talk about genetic testing, the importance of spending time together, differences in coping between parents, feelings of guilt and forgiveness, and worries about the future health of the family. This research highlights that genetic testing impacts families deeply and that healthcare providers should address these family-level challenges to better support families during an already difficult time.

## 1. Introduction

Recent studies in Western countries indicate that a growing proportion of children with cancer harbor a cancer predisposition syndrome (CPS), estimated at 10–15% or higher depending on the sample studied [1,2,3,4,5]. The possibility of diagnosing a CPS in pediatric cancer is highly relevant for treatment adaptation, future risk prediction, surveillance, consideration of predictive testing for at-risk relatives, and family planning [6]. Consequently, germline genetic testing is progressively being integrated into standard pediatric oncology care in different parts of the world (e.g., Belgium, The Netherlands, Canada, Australia, USA, etc.) [7,8]. Despite the medical benefits, offering germline genetic testing also presents families with notable challenges. In addition to ethical considerations (such as balancing a child’s best interest with their right not to know) [9,10,11,12,13], the psychological impact on the patient and their family can be significant.

First of all, facing childhood cancer is already a life-threatening stressor and family stress models assume that a stressor affecting one family member also affects the other family members and the entire family system [14]. Empirical evidence supports this theoretical perspective. Next to pain, fatigue, and reduced immunity, the ill child often endures anxiety and uncertainty [15]. Siblings frequently report a diminished quality of life and adverse emotional reactions [16] and, in comparison to parents of healthy children, parents of children with cancer often report significantly higher levels of distress, posttraumatic stress symptoms, parental conflicts, emotional difficulties, and physical complaints [17]. Additionally, families—as a whole—often experience altered roles and responsibilities and increased conflict, but also increased cohesion, communication, and family support [18].

Second, the introduction of germline genetic testing into pediatric oncology care may represent an additional stressor affecting these families, namely the possibility of identifying an underlying predisposition to cancer. On the individual level, recent reviews have shown that ill children and their parents experience a wide range of emotions as a consequence of germline genetic testing, including depression, anxiety, distress, uncertainty, and loneliness [19,20]. Although these negative emotions can resolve in the long-term and can co-exist with feelings of empowerment and positive emotions like relief and hopefulness, several individual risk factors (e.g., female gender, having pre-existing mental health issues, and a lower education level) and contextual risk factors (e.g., families receiving different genetic test results across children or children witnessing distress or illness in a parent) have been identified that can make the impact of germline genetic testing more emotionally taxing [19,21,22,23,24,25]. This psychological impact should be carefully considered in oncology care and genetic counseling [26]. In addition, germline genetic testing in childhood cancer likely also generates a broader family-level impact. Especially since a germline predisposition may affect families by influencing their health outcomes (including susceptibility to cancer). Indeed, the results of germline genetic testing for a CPS can have direct and significant implications not only for the affected child but also for parents, siblings, and extended relatives, who may be found to share the same genetic predisposition [7]. The family-level impact of germline genetic testing in childhood cancer, however, remains understudied, so a detailed and systematic examination of shared experiences in families navigating the complexities of childhood cancer and associated germline genetic testing is needed to organize comprehensive support for them. Therefore, the current study aimed to achieve an in-depth understanding of how germline genetic testing in childhood cancer impacts families.

## 2. Materials and Methods

Focusing on the family as a whole, the parents as a couple, and the parent–child relationship as units of analysis, Multi Family Member Interview Analysis (MFMIA) [27] was used as a guiding methodological framework in this qualitative study. One-to-one in-depth semi-structured interviews were conducted with both parents separately. This allowed each parent to provide their own perspective [28] without having to factor in their partner’s feelings [29]. Children were not included in the interviews due to their health status and/or age.

### 2.1. Participants

Interviews were conducted with both parents in six families who opted for and received the results of extensive germline sequencing for childhood cancer predisposition in a diagnostic setting. The children received three possible outcomes: diagnosis of a cancer predisposition syndrome (CPS) linked to the malignancy they developed (N = 2), diagnosis of a secondary finding (a secondary finding is a (likely) pathogenic variant that is not causative of the patient’s presenting condition but that might have implications for the patient’s future or for the wider family. In this case the reported secondary findings are all cancer-related actionable, late-onset (likely) pathogenic variants.), i.e., a genetic variant that could not be linked to the malignancy they developed (N = 2), or no genetic diagnosis, i.e., no (likely) pathogenic genetic variant was identified (N = 2). All families were Caucasian, consisting of four Belgian and two Dutch families. The parents were aged between 30 and 50 years old and had attained a middle or higher level of education (educational level was identified as lower (i.e., did not complete secondary education), middle (i.e., completed secondary education but did not pursue higher education), and higher (i.e., completed education beyond secondary level)). More detailed characteristics of the families can be found in Table 1. Approval from the medical ethical committee of Ghent University (Hospital) had been obtained (BC-08213-AM04), and informed consent forms were signed by all participating parents.

### 2.2. Data Collection

This study is part of the larger ongoing Belgian DHECIPR project (Diagnosing HEreditary predisposition syndromes for Childhood cancer: Implementation in clinical Practice) where extensive germline genetic testing was offered to all patients (0–18 years) diagnosed at the Department of Pediatric Haemato-Oncology of Ghent University Hospital (Belgium) between 2021 and 2025. Depending on their maturity, children were involved in the genetic counseling and, if 12 years or older, they also signed a simplified informed consent form to initiate germline genetic testing. In the informed consent process, parents (and children if 12 years or older) agreed to receive information about all (likely) pathogenic variants related to the disease and actionable secondary findings. This was thoroughly discussed during genetic counseling prior to the testing. Psychological support was available for families interested in germline genetic testing both during consultations and between sessions, should families need it. The germline genetic testing was performed by exome sequencing followed by data analysis focused on a panel of 197 genes associated with a childhood cancer predisposition. The families who opted for germline genetic testing were invited for discussion of the results in a multidisciplinary consultation (with a clinical geneticist with large expertise in oncology, a clinical psychologist, and a pediatric hemato-oncology nurse specialist) at the center for medical genetics. The psychosocial component of the DHECIPR project, consisting of a questionnaire study for both parents and the child (≥12 years) and an interview study (only parents), was offered in parallel with the medical component of the study (i.e., collecting clinical and genetic data to prospectively validate the MIPOGG referral tool [30] in real-world clinical practice). Interviews took place after consent and disclosure of the genetic results. Exclusion criteria were (a) insufficient proficiency in Dutch and (b) prior confirmation of a genetic cancer predisposition in the patient.

Of all nine families who received a diagnosis of a CPS or secondary finding (N = 6 + 3), the parents were subsequently invited by the psychologist to complete an interview about the impact of the germline genetic testing on their family. In five families, one or both parent(s) preferred not to participate in the interview study, citing lack of time and/or interest as primary reasons. The remaining four families (N = 2 + 2), where both parents agreed to participate in the interview study, were included. Of the families where no underlying genetic variant was discovered, two families were randomly selected and contacted by the psychologist; both families agreed to participate. Interviews were conducted with each of the participating parents individually and were held at the patient’s home, at the University Hospital, or via video conference. To avoid influencing each other, both parents were interviewed consecutively without having contact, except for one couple. All interviews were conducted by the same interviewer (S.V.H.: a female clinical psychologist with experience in oncology, family therapist in training, and PhD student with experience in qualitative research), were audio recorded, and lasted between 19 and 68 min. Verbatim transcripts of these interviews served as the raw data for this study. The interviews were based on an interview guideline (see Appendix A; developed by S.V.H.; S.H., a female clinical psychologist in the Center for Medical Genetics, family therapist, and researcher with experience in qualitative research; M.V.S, a female clinical psychologist and researcher with experience in qualitative research and pediatric oncology; and L.V., a female clinical psychologist, family therapist, researcher, and professor in Couple and Family Psychology, and based on literature review, other interview guidelines within pediatric oncology [31], and clinical experience) and consisted of open-ended questions about (a) the experience of the germline genetic testing, (b) the impact on the parent as an individual, (c) the impact on the family relationships, (d) the impact on the parents’ romantic relationship, and (e) the impact on their environment. Participants’ experiential accounts were facilitated by prompts, in order to encourage the participants to give personal accounts [32]. Pseudonyms were used to protect participants’ anonymity.

### 2.3. Data Analysis

Inspired by Interpretive Phenomenological Analysis (IPA) [32] and Dyadic Interview Analysis [28], MFMIA [27] aims to understand broader family dynamics by obtaining and combining the perspectives of multiple family members [33] and enables a detailed and systematic examination of shared family experiences [27,34]. Data were analyzed from a post-positivist position. The central focus is on the meanings that these experiences have for the families and not to produce objective data. To ensure a robust analysis, all phases were completed by two authors (S.V.H. and S.H.), with consensual agreement [35], i.e., working collaboratively to co-construct the best representation of the data by integrating multiple perspectives and resolving coding differences through discussion until consensus is reached.

In the first phase, individual interviews were analyzed separately following IPA principles. S.V.H. and S.H. both analyzed 100% of the interviews and independently read each transcript multiple times to familiarize themselves with the participant’s account, then annotated the transcript with initial observations. These observations (e.g., “this mother wants to spare her child’s future children from cancer”) were then translated into broader themes (e.g., “preventing cancer in future generations”). Parallels between emerging themes were explored, leading to a clustering of themes for each case. This process was repeated for each participant. In the second phase, themes relevant for each parental dyad were identified by combining the themes of both parents after analyzing their individual transcripts. In the third phase, parallel themes between parental dyads from different families were identified and discussed until consensus was reached among two additional authors (L.V. and M.V.S.). This process of investigator triangulation allowed enhancing the validity of the study [36]. The final list of subordinate (e.g., “preventing the inheritance”) and superordinate (e.g., “concerns regarding the other children”) themes reflected patterns of convergence across different dyads, based on unique aspects of each parent’s and couple’s experiences. Finally, all themes were translated into a written account, elaborating on the analysis and illustrating it with direct quotes from the participants.

## 3. Results

### 3.1. Childhood Cancer Versus Germline Genetic Testing

When parents were asked about their experiences with the diagnostic germline genetic testing, they predominantly characterized it as a beneficial component of their child’s oncological treatment process, expressing appreciation for its implementation.

“For us, it is actually a kind of given. Look, a tumor isn’t there for no reason. There might be a cause somewhere, and then it is good to start early to trace the cause or a possible cause. So, in that sense, we were glad that it was addressed quickly.”(Father, Family 1)

However, when explicitly asked to describe the impact of the germline genetic testing on their families, parents encountered difficulties in articulating its specific impact. The challenges associated with the cancer diagnosis and subsequent treatment often overshadowed those related to germline genetic testing or the impact of the germline genetic testing was indistinguishable from that of the cancer diagnosis. This resulted in the emergence of an overarching theme: childhood cancer diagnosis versus germline genetic testing. This was consistently evident across all other identified themes.

“Yeah, you know, from the moment of the diagnosis, you are so overwhelmed that you are not even thinking about it [the germline genetic testing]. You are so consumed every day with going to radiation treatments, you are very busy with everything (…), it really fades into the background and I know it’s important, but it’s more important to focus on the here and now for a while.”(Mother, Family 1)

“It’s very difficult to say that it [the experienced changes in the family] is really linked to that genetic result. Because, as I said, it came just a week after the chemo. So, for us, that genetic result is kind of in the same bag. That’s just how it is, but our family has changed definitely.”(Mother, Family 4)

Despite the challenge of pinpointing the unique impact of germline genetic testing, parents’ narratives clearly revealed impact on their families associated with the germline genetic testing. This impact was grouped into five main themes: (a) talking about germline genetic testing, (b) being together matters (more), (c) differences in coping with germline genetic testing between partners, (d) feelings of guilt and mutual forgiveness, and (e) concerns about other the future health of the family (see Figure 1).

### 3.2. Theme 1: Talking About Germline Genetic Testing

Subtheme 1a: Transparency with children. When offered germline genetic testing, parents found it important to maintain transparency with their children about the germline genetic testing process. They believed that their ill child, as well as the siblings, should be informed about the possibility of testing and that they should be included in discussions if the children were of an appropriate age. In addition—as some children may exhibit more interest than others—parents adjusted their communication to meet each child’s needs, providing more information to those who sought it while not overwhelming others.

“We also told Jules (Pseudonyms were used to protect participants’ anonymity.) and the sisters about it. The oldest, for example, is studying pharmacy, so she is very interested in the whole process and how it affects Jules. (…) We explained it to them. The middle one wants less to do with it. She is sixteen years old and is also less interested in it.”(Mother, Family 5)

Families valued open conversations with their children without imposing pressure to participate, especially when discussing germline genetic testing with the affected child. There, parents strived for a balance between informing and supporting their children to reach a shared decision.

“We talked a lot. It’s really difficult because she is still a child and can’t really make drastic decisions, but this is about her and her life. And she has to go through all of it. So, I definitely believe that if anyone deserves to have a say or a choice, it is her. And it’s not about ’oh, what do you want?’ and then ’ok, we’ll do it that way’… No, we talk about it.”(Mother, Family 4)

Even though parents considered shared decision making important and supported their children’s autonomy, parents did experience relief if their children’s preferences align with their preferred course of action.

“We also talked about it with the rest of our children because the doctor wanted to have them tested as well. At first, I didn’t want it, but I left the decision up to them. Fortunately, they felt the same way and didn’t want to know about it right now.”(Mother, Family 1)

Subtheme 1b: Protecting Each Other. Discussions regarding germline genetic testing within families were conducted with sensitivity, as both parents and children strived to protect one another. Parents carefully selected both the content and manner of information shared with their children.

“We discussed more about how we are going to explain it to the children. What are we going to say, what can we not say? What is the best way to phrase it, to talk about the genetics?”(Mother, Family 3)

Parents felt ambivalence and aimed to balance honesty with the desire to protect their children’s emotional well-being. They prioritized maintaining a carefree childhood for their children. Indeed, given the significant challenges their child had already faced due to the cancer diagnosis, some parents aimed to shield them from additional stress by selectively withholding information or postponing discussions about the testing.

“We always decided to let Oscar know as little as possible about everything. Because he had to go through so much with the chemo, we tried to take away as much of the emotional burden as we could for him. So, he knows that he is sick, he knows what he has, but that’s it.” (Father, Family 6)

Another way of protecting was by maintaining a strong “front”. Parents frequently described suppressing their own worries to prevent placing an additional emotional burden on their children.

“The emotions are intense (…) [starts crying], but you still want to be strong for your family, for your child.”(Mother, Family 1)

Likewise, parents reported that their children also tried to take on a supportive role by being positive in order to protect their parents from becoming even more distressed.

“When Emma would sit with me, she would say… ‘Mom, are you okay?’ And she would look at me and say… ‘Mom, it will be alright. Everything will be alright. Now we know, and it will be all right’.”(Mother, Family 4)

Subtheme 1c: Confrontation with the extended family. Parents reported that their extended family members were sometimes concerned about the (results of the) germline genetic testing, regarding the consequences for themselves and their own children. While parents understood the concerns of the extended family, they often were reluctant to be too open and found themselves unsure about how and what to communicate regarding the testing. Therefore, some parents chose to withhold information about the germline genetic testing from their extended family until the results were known to prevent unnecessary worry.

“I didn’t go around telling everyone: ‘we’re going to get tested because it could be genetic’. That would only make more people worry. We were going to tell them when the time came and we knew the results. (…) For example, my brother is someone who immediately thinks it will happen to him too. And if I said it could be genetic, he would start worrying that his daughter and son might have it as well. He wouldn’t take it well. So, I prefer to protect him rather than make him worry as, for now, there’s nothing to worry about.”(Mother, Family 6)

Besides wanting to avoid potential unnecessary worry in the extended family, another reason for not sharing information was to protect their nuclear family. Some parents preferred to focus on their own child and family without the additional emotional strain of dealing with the concerns and reactions of relatives. By withholding information, they shielded themselves and their nuclear family from potential emotional stress.

“That also opens a whole can of worms regarding the family, instead we can now focus only on this and not worry about the entire family potentially being involved.”(Mother, Family 1)

### 3.3. Theme 2: Being Together Matters (More)

When confronted with a cancer diagnosis and a genetic predisposition, parents reported that their family experienced a shift in values. They placed greater importance on family time and invested more effort in spending moments together. Parents perceived that the shared experience of navigating cancer and germline genetic testing made their families more appreciative of the time they have together. These moments of togetherness had become a source of strength and comfort, reinforcing their bonds and highlighting the importance of prioritizing family over other obligations.

“I’m not saying we leave everything; we keep going, but now we’re more likely to say, ’Does it have to be this week? There’s always next week.’ We say, ’We can also do something fun with the kids.’ (…) There was a movie night at school, and this is the first year we’ve attended. Now we say, ’Yes, we’re going to do that, and we’re going to go as a family.’”(Mother, Family 3)

### 3.4. Theme 3: Differences in Coping with Germline Genetic Testing Between Partners

As a couple, some parents dealt with the germline genetic testing in a similar manner, finding alignment in their response to the ongoing germline genetic testing and the subsequent result. However, other couples managed the experience differently. Typically, one partner assumed the role of the “practical level-headed one”, approaching the germline genetic testing rationally, searching for a solution, and feeling reassured by receiving adequate information. Conversely, the other partner took on the role of the “emotional thinker,” approaching the testing based on feelings, ruminating about potential negative outcomes, and not finding reassurance in information alone. Partners described being very aware of their differences and trying to respect each other’s approaches.

“It did bring a kind of burden with it. Lisa (partner) was a bit more afraid of the result, but in the end, I am just a bit different from Lisa. Lisa would be more likely to say, ’What if,’ whereas I would be more likely to say, ’If we know, we can respond to it.”(Father, Family 3)

“For me, it faded into the background and I thought, we’ll worry about it when we get the results. You can worry about it now, but that doesn’t make any sense. Well, I’m the type of person who moves on quickly and we’ll see when it comes. I know that Dave thinks about it and worries about it much more… But he’s not a talker, so…”(Mother, Family 6)

Parents also tended to make assumptions and fill in for each other based on what they observed in their partner’s external behaviors. In this way, they assigned roles to each other, where differences in coping seemed to be amplified.

“I am the one who is more emotional. (…) And Bert (partner) is a bit like my dad; he only looks at the positive things. So, if the doctor says, we can do this, we can do that, that is their bit of reassurance. I don’t mean to say that he isn’t afraid, but he doesn’t really think about: oh, what could happen? Or at least, he doesn’t talk about it.”(Mother, Family 2)

Partners may also be considerate of each other’s emotional states and intentionally withhold certain thoughts to avoid causing distress or arguments due to their differences in dealing with the germline genetic testing.

“That is not something I want to talk about with someone who is emotional. I haven’t actually talked about it with anyone except you and the psychologist. I think that I am too dry in my thinking about it. And I don’t want to discuss it with anyone, especially not with Valerie (partner). (…) I also don’t want it to cause any arguments because of what I think.”(Stepfather, Family 4)

“You say… I am very rational and very strict in this. On the other hand, it also visibly affects you when speaking about it, doesn’t it?”(interviewer)

“It, it, it definitely affects me. Uh… [gets emotional] But in my mind, there is a solution.”(Stepfather, Family 4)

### 3.5. Theme 4: Feelings of Guilt and Mutual Forgiveness

Subtheme 4a: Transmission and survivor’s guilt. Regardless of the test results, the theme of transmission guilt consistently and spontaneously surfaced in all conversations with parents. Parents expressed concerns about the possibility of having transmitted a (likely) pathogenic variant to their offspring, highlighting the emotional burden and feelings of responsibility for their child’s genetic condition.

“Sometimes you think, I just hope it has nothing to do with the known syndrome [that already runs in the family] (In this family some family members (and not the patient) were known carriers of a predisposition. At the start of the study the carrier status of the child was not known, but the genetic analysis revealed this.) because I would really find that terrible. I mean, otherwise, it comes from me.”(Mother, Family 1)

Moreover, according to some parents, grandparents also shared this burden, feeling responsible for the cancer predisposition syndrome manifesting in their grandchild. This (inter)generational aspect highlights the broader familial impact of the possibility of a genetic predisposition and the accompanying emotional burden.

“My mother takes this very much to heart; she’s quite affected by it because she, of course, had surgery herself at one point. Fortunately, it always remained just that one tumor, but she is really upset that the youngest grandchild is indirectly affected by what she had. While she knows, rationally, that she can’t feel guilty about it, she still feels it somewhere, yes.”(Mother, Family 2)

“That was ultimately a big relief for me. I also started to cry because I could reassure my mom that it didn’t come from her side.”(Mother, Family 6)

When germline genetic testing failed to identify a cause for the tumor, many parents initially expressed relief from their feelings of guilt. However, a persistent sense of guilt often remained.

“It is now clear that something went wrong due to nature and that none of us could do anything about it. We just have to get through it. It was not his fault, or our fault, or my fault that he got it. That is life, but it doesn’t mean I feel any less guilty that he got it.”(Mother, Family 6)

Additionally, some parents who did not carry the (likely) pathogenic variant themselves also experienced a form of guilt, often referred to as survivor’s guilt. This was characterized by a deep sense of injustice and, in some cases, a desire to share the burden with their affected child.

“I was so angry that Emma had to bear this alone. It’s strange to say, but I felt disappointed that it was her again and not me. They’ve tested me too. I would have preferred to have it myself, along with my two children. Not that I want them to have it, but it was just Emma again…”(Mother, Family 4)

Subtheme 4b: Exonerating each other as parents. At the individual level, feelings of transmission and survivor’s guilt were prominently felt. However, at the interpersonal level, mothers and fathers consistently demonstrated mutual forgiveness. This understanding was frequently expressed even before receiving germline genetic testing results.

“We had already said, ’don’t be angry if it comes from me, okay.’ But then one says, ’well why? You didn’t know there was going to be anything either.’ My parents wouldn’t have known that they passed it on, and it’s been in the genes for generations.”(Mother, Family 3)

“Well yes, Lisa also had a bit of the mindset of ’if there’s anything, then it will come from my side of the family.’ But yes, in the end, it’s just the bad luck that it came from her side; just as easily, it could have come from my side.”(Father, Family 3)

Overall, in nuclear families, parents’ primary concern was the health of their child, rather than determining which partner was to blame. They emphasized unity and actively avoided placing blame on one another.

“You are dealing with your son’s illness. I mean, Jules needs to get through it, and where it comes from—whether it’s from me or my wife—we’re not going to start saying it’s your fault or my fault.”(Father, Family 5)

In contrast, the dynamics in divorced families may differ significantly.

“I was so angry. I thought, that man has brought nothing good into her life. Since she was born, he has only brought us misery. I mean, and that’s actually not fair, because he probably can’t help it. This might be the only thing he can’t help. And afterwards, when it all calms down, you know that, but at that moment…”(Mother, Family 4)

### 3.6. Theme 5: Concerns About the Future Health of the Family

Subtheme 5a: Potential risk for siblings and children in the extended family. Gradually, or sometimes simultaneously with their child’s cancer diagnosis, parents began to express concerns about the potential cancer risk for their other children, the children of their family members, and themselves.

“We actually had that [question] at the time of diagnosis, yes: ’Is it something hereditary? Are we passing on that tumor ourselves? Should we already be looking to check for ourselves or for his brother?’”(Mother, Family 3)

“Our other son and also within the family. We have more children, more people. So, if there is something genetic, it doesn’t stay just within our immediate family. Then, of course, it also affects the extended family, and that can be quite extensive.”(Father, Family 6)

Subtheme 5b: Future offspring of the ill child. Parents were not only concerned about the current health of family members but also contemplated the future health of their potential grandchildren. They reflected on whether prior knowledge of genetic risks might have influenced their decisions to have children. Conversely, some parents recognized the challenges but expressed gratitude for modern medical interventions that could help future generations manage these risks.

“I find it very difficult that he is also a carrier. Then you know that the syndrome doesn’t just stop, it continues in their lives, and now he already knows it. Yes, I sometimes say that if I had known this beforehand—especially if this tumor had been related to this syndrome—I would never have had children, but I didn’t know. He can now make the choice, you know, to not have children, and I find that quite intense at his age.”(Mother, Family 1)

“If our children want to have children later, then it will have to be with all sorts of modern medical interventions. I find that a shame for them, but on the other hand, it’s also something very beautiful that it is possible.”(Mother, Family 2)

In addition, parents also seemed considerate of the impact it would have on their child’s future partner.

“It will of course be a bit strange for the children in the long run, because if they have a partner they want to be with, then at some point they will also have to say: well, if we want children, it will never happen in the normal way.”(Mother, Family 2)

## 4. Discussion

This study is, to our knowledge, one of the first to explore how parents experience the family-related impact of germline genetic testing offered as the standard of care in the context of pediatric cancer. Genetic information can affect multiple family members, and adopting a family-wide approach helps us to identify the psychological impact of germline genetic testing on the entire family system.

While germline genetic testing was generally viewed as a valuable and straightforward step in the oncology trajectory, it seems that parents found it difficult to distinguish its impact from the overwhelming experience of their child’s cancer diagnosis and the current treatment, which is their primary concern. This is in line with the findings of McGill et al. [37], where the parents of children with cancer described genetic consultations as a secondary concern to the immediate stressors of their child’s treatment. This highlights the complex emotional landscape in which families navigate both the immediate challenges of the cancer and the potential long-term implications of germline genetic testing, such as the increased future cancer risk in the child, the increased cancer risk in family members, and the lifelong, intensive surveillance in carriers. An interesting finding from our study is that, despite the overarching stressors related to cancer treatment, parents recognized the role of germline genetic testing in influencing family dynamics such as discussing testing, the importance of togetherness, variations in how partners manage testing, feelings of guilt, and concerns about their other children and family members. These findings provide a nuanced understanding of how families deal with germline genetic testing and its broader implications and differ from previous more descriptive findings by focusing on processes rather than on outcomes [8,9,25,38].

Family communication plays a vital role throughout the germline genetic testing process in cases of childhood cancer. This “*talking about testing*” encompasses not only the disclosure of genetic results but also the pre-test discussions that shape the decision-making process. Our data show that, within nuclear families, parents often emphasize maintaining an open dialogue with their children, encouraging *“transparency with their children”* and promoting shared decision making. However, the inherent emotional complexities of such conversations frequently introduces challenges. The delicate balance between fostering transparency and the instinct to protect one another from psychological distress—referred to as “*protecting each other”*—becomes a central tension. The desire for emotional protection within families aligns with established psychological constructs such as protective buffering [39] and double protection [40]. These concepts describe the tendency of individuals to withhold personal thoughts, feelings, or concerns in an effort to protect a loved one from emotional distress. In the case of double protection, both parents and children engage in this mutual shielding, attempting to spare each other from additional emotional burden. This tension between transparency and protection is particularly pronounced in the context of genetic testing, where the stakes are both emotionally and medically high. Family studies highlight the importance of open communication in coping with illness, noting that transparency can foster collective resilience and meaning-making within the family unit [41]. However, as Afifi et al. [42] emphasize, individuals often face competing desires for information and avoidance when the emotional cost of full disclosure feels too high. This sheds light on the ambivalence parents and children experience in genetic testing discussions, where the imperative for transparency is counterbalanced by the urge to protect. It seems crucial to address this felt contradiction within families in order to enhance shared coping and meaning making of the challenges of germline genetic testing [43].

Additionally, our findings highlight the involvement of *extended family members* in conversations about germline genetic testing in the context of childhood cancer. Extended family members often express both an interest in the germline genetic testing process and concerns about the health implications for the ill child, as well as the potential risks for themselves and their offspring. Similar to communication within nuclear families, our results suggest that discussions with the extended family require a careful negotiation between transparency and emotional protection. On the one hand, initiating conversations with extended family members before receiving germline genetic test results may provide them with an opportunity to process potential outcomes, engage in informed discussions, and offer emotional support to the nuclear family [44]. This approach aligns with findings from studies on family-centered communication, which suggest that early, transparent discussions about genetic risk can reduce uncertainty and enhance coping mechanisms across familial networks [45]. On the other hand, parents are often cautious about the emotional impact such disclosures may have on extended relatives. Parents in our study felt a responsibility to assess the emotional capacity of extended family members before broaching the subject of germline genetic testing. They aimed to assess the emotional readiness of others before sharing sensitive information. Common reasons for delaying discussions about sequencing with extended family members include the relatives’ potential anxiety about their own health risks, as well as the emotional toll of navigating a loved one’s illness [44]. Some parents therefore prefer to wait until they receive results before disclosing their decision to undergo germline genetic testing, as a way to avoid imposing unnecessary emotional strain on relatives before any concrete information is available.

Furthermore, the theme of “*being together matters (more)*” reflects a shift in (prioritizing) certain family values, often triggered by the experience of a cancer diagnosis [31], but potentially intensified by the experience of a genetic risk diagnosis. Our data show that this shift towards (more) togetherness can be driven by a heightened awareness of the uncertainty surrounding the future. In such situations, families may re-evaluate their priorities, placing greater importance on family-centered activities that reinforce emotional connections. These shared experiences seem to strengthen family bonds, underscoring the importance of time spent together as a source of resilience [46]. However, this shift may not be universal. Other studies have shown that parents of children with cancer predisposition syndromes and the children themselves sometimes report feelings of alienation from other family members. The rarity of such a diagnosis can create a unique emotional and social divide within the family [24,37]. Thus, while a genetic diagnosis can foster family closeness for some, for others, it may lead to a sense of loneliness.

Another prominent theme we identified in the context of germline genetic testing for childhood cancer was the tension between *emotional responses and rational decision making*. In the decision to initiate germline genetic testing, couples typically demonstrated a preference for knowing rather than not-knowing and prioritized the potential medical benefits over fear-driven reactions, arriving at a joint decision grounded in the perceived utility of the results. This principal attitude toward germline genetic testing seems consistent with the previous literature [25,47,48]. In addition, families generally experienced little burden in opting for germline genetic testing [25,49]. However, alignment in decision making does not necessarily translate into a shared coping strategy throughout the germline genetic testing process. Our findings show that one partner may focus more on practical solutions and the other on the emotional ramifications. This aligns with previous studies showing that—even when confronted with the same stressor—coping strategies (e.g., problem-focused versus emotion-focused) can differ significantly between partners [50]. Emotion-focused coping aims to regulate emotional responses during stressful situations, while problem-focused coping targets addressing and altering the source of stress. Problem-focused coping tends to be more appropriate in situations that are changeable (i.e., controllable), whereas emotion-focused coping is more useful when facing inevitable or uncontrollable events [51]. In the context of germline genetic testing, the effectiveness of these strategies seems complex to determine; as stated by Fantini-Hauwel et al. [52], on the one hand the test results may seem uncontrollable due to inherent factors, while on the other hand families can exert some control by engaging in regular screenings to manage their risk. When facing germline genetic testing in childhood cancer, these divergent coping styles become more pronounced as couples have to navigate the psychological complexities of the process together. Here, our data demonstrated that both partners strived to respect one another’s approach; sometimes this divergence led to a more balanced way of coping, where partners learned from each other, and other times this divergence also led to unspoken emotional burdens, as some parents withheld their concerns or thoughts to protect their partners or children from additional stress or conflict. These findings emphasize the need for healthcare professionals to offer tailored support to both parents to address emotional and informational needs more comprehensively.

Our results also revealed the pervasive theme of *guilt*, manifesting in concerns over the transmission of (likely) pathogenic variants and feelings of survivor guilt—where parents spared from the (likely) pathogenic variant experience guilt on behalf of their affected child. This finding aligns with other research on hereditary oncologic conditions, which demonstrates that parents often feel a profound sense of responsibility and guilt, despite the genetic risk being beyond their control [19]. Interestingly, parents in our study often preemptively exonerated each other from *blame*, reflecting a mutual understanding of the uncontrollable nature of genetic inheritance. However, the experience of one family, struggling to navigate these emotions, indicates that not all families are able to provide mutual support, suggesting that certain families may require additional psychological support.

Another significant finding was the persistent *concern about the future health of the family*, reflecting the far-reaching emotional and practical implications of germline genetic testing. Parents in our study expressed apprehension not only about the risks faced by their affected child but also regarding the potential impact on their other children, future grandchildren, and extended family members. Indeed, other studies show that gaining knowledge about whether other family members could be at risk and implications for their and their child’s future family planning were seen as advantages of testing [48,49]. These findings highlight the intergenerational ripple effect of germline genetic testing and underscore the need for family-centered genetic counseling to address ongoing and future concerns, taking into account the availability of resources and time. Additionally, our results demonstrate that parents communicate little about their own genetic risk. During the germline genetic testing process, they primarily focus on their role as caregivers, rather than viewing themselves as individuals at risk. As a result, they either do not address their personal concerns or choose not to discuss them. This highlights the complexity of parental perspectives, as their focus on caregiving often takes precedence over acknowledging or discussing their own genetic vulnerabilities.

### 4.1. Clinical Implications

Given the complex emotional dynamics that families experience during germline genetic testing in the context of childhood cancer, there is a clear need for family-centered counseling that goes beyond discussing the individual risk of the ill child. Healthcare providers (e.g., geneticists, but also psychologists and genetic counsellors) could therefore gather the involved family members and facilitate discussions that address both immediate medical concerns and the psychological impact on the entire family system. This includes (a) helping parents navigate discussions with their children, (b) preparing nuclear families for conversations with extended family members, (c) discussing feelings of guilt by normalizing the uncontrollable nature of genetic inheritance and promoting mutual understanding among families, (d) exploring couples’ differences and similarities in coping, so partners and clinicians can discover if they have converging or diverging needs and how these can be met, and (e) acknowledging the future-oriented concerns by discussing the preventative health strategies for future generations. This approach could provide a supportive space for families that can foster familial coping and shared understanding.

### 4.2. Study Limitations and Suggestions for Future Research

A key limitation of this study is the need for caution when interpreting the impact of germline genetic testing on families in the context of childhood cancer, as our findings indicate that this impact is intertwined with the impact of the cancer diagnosis, suggesting that the results should be understood within the broader context of families dealing with the challenges of a cancer diagnosis. Additionally, the sample size was relatively small, which may limit the diversity of family dynamics observed. A larger sample could provide a more comprehensive picture, could potentially uncover additional themes, and could explore differences in the experienced impact of germline genetic testing based on when it was offered to families. Another limitation is that the sample did not include families with children who have a de novo variant. Including such families could help researchers identify whether the emotional and communicative dynamics differ when the risk is not linked to one parent’s genetic material. Furthermore, from parents’ perspectives, the germline genetic testing process seems to resonate with various family members, which highlights the potential value of future research involving other family members. First, future studies could focus on how to involve children, siblings, and extended family members in research, as their experience of the impact of germline genetic testing on their family may differ from parental views. Second, future studies could explore how to involve extended family members in clinical practice, as it would add to our knowledge and understanding of family dynamics (e.g., the benefits and challenges of family communication about germline genetic testing prior to disclosure of the test results). Finally, it would be of interest to develop guidelines to support clinicians in addressing the described themes with families.

## 5. Conclusions

Having a child being confronted with a life-threatening cancer diagnosis often makes it difficult for parents to recognize the specific impact of germline genetic testing on their family. However, as healthcare providers engage in more in-depth discussions, a range of important issues tends to come to light. When confronted with germline genetic testing, families encountered complex emotional and communicative challenges. The decision to undergo germline genetic testing sparked various discussions within families, as parents strived to balance transparency, protection, and consideration for their nuclear and extended family. Germline genetic testing also triggered concerns in families about the health of siblings, future offspring, and extended family and made them place greater value on being together. While it elicited experiencing guilt, parents also showed mutual forgiveness and, despite their differences in coping, tried to respect each other’s approaches in dealing with the germline genetic testing. The emotional complexities surrounding germline genetic testing for childhood cancer underscore the need for nuanced approaches to family communication by geneticists, genetic counselors, and other involved clinicians (e.g., pediatric oncologists). A deeper understanding of the protective mechanisms families employ, along with interventions designed to encourage shared meaning-making, will be critical in enhancing the psychological well-being of the family as a whole. Future research should continue to explore how families navigate these emotional dynamics, particularly in the context of emerging genetic technologies and evolving ethical considerations in pediatric oncology.

## Figures and Tables

**Figure 1 cancers-17-00517-f001:**
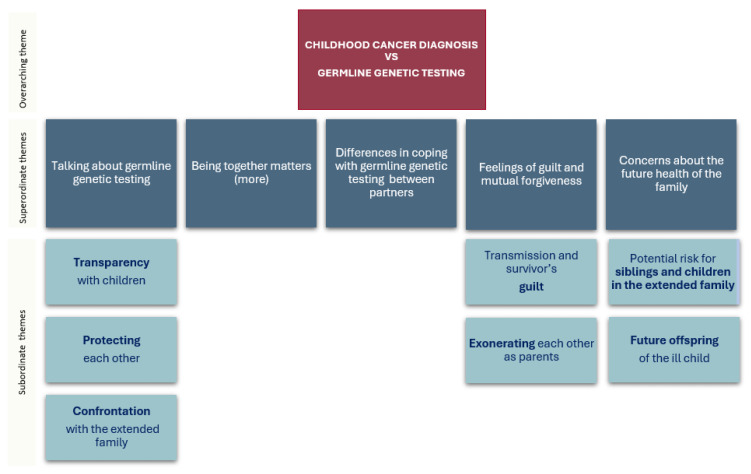
Impact of germline genetic testing: superordinate and subordinate themes.

**Table 1 cancers-17-00517-t001:** Characteristics of participating families.

Families	Both Biological Parents Participated	Gender of Ill Child	Age of Ill Child	Time Since First Cancer Diagnosis	Number of (Half-) Siblings (+Carrier Status)	Sequencing Result of Ill Child
Family 1	Yes	Male	12 year	8 months	X (not known)	CPS found
Family 2	Yes	Male	11 year	3 years (This patient developed several metachronous primary tumors, with the most recent 4 months prior to the interview)	1 (carrier)	CPS found
Family 3	Yes	Male	5 year	6 months	1 (carrier)	Secondary finding
Family 4	Mother and stepfather (In this particular family there was no longer any contact with the biological father, and the stepfather had been fulfilling the father role for many years.)	Female	14 year	5 months	2 (no carriers)	Secondary finding
Family 5	Yes	Male	12 year	9 months	2 (N/A)	No CPS found
Family 6	Yes	Male	4 year	1 year	1 (N/A)	No CPS found

## Data Availability

Data are contained within the article or Supplementary Material. The original contributions presented in this study are included in the article/Supplementary Material. Further inquiries can be directed to the corresponding author(s).

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
