# Peer review of "Family-Level Impact of Germline Genetic Testing in Childhood Cancer: A Multi Family Member Interview Analysis"

_cancers, 2025, doi:10.3390/cancers17030517_

Round 1

Reviewer 1 Report

Comments and Suggestions for Authors

Introduction/Methods:

Page 3: Regarding the term ‘Incidental finding’, I would prefer ‘secondary finding’ as it is not incidental when you perform WGS and analyze the data.

Page 3: ‘No genetic variant was identified’. I think you should change to ‘No pathogenic germline variants were identified’, as I am sure numerous genetic variants was found?

Page 4: I was wondering whether the authors have ensured that data saturation was reached? I only states that they offered interviews to all families with pathogenic/secondary findings, and that two families with no pathogenic germline variants were invited per random. Could you elaborate on this?

Page 3-4: Would it be possible to state a little about the counseling guidelines regarding the children. When we reach the results section it says that parents make the choice of being open about the process with children and siblings, but you must also have some guidelines as to how pre-teens and adolescents are counseled? Could be nice to know, as there are both 12-14-year-olds included in the study.

Results:

Page 7: I found the subtheme of ‘1c: Confrontation with the extended family’ extremely interesting. And this is also new knowledge that adds to our understanding of family dynamics i.e. that the larger family dynamic is somewhat of a potential minefield (which is often different in families with no active disease going through genetic testing and counselling, where sharing seems easier (in many families). If you have more data on family dynamics and how your interviewees handled this, please consider including it in the manuscript.

Page 8: earlier publications have found the just the knowledge that genetic sequencing is possible brings in itself a burden, because now you cannot know that this is a possibility. I would ask the authors to including that in the discussion regarding theme 3.

Page 8: I seem to recall (from the methods section) that you had excluded families with known cancer predisposition syndromes? But in page 8 ‘Mother, Family 1’ states: "Sometimes you think, I just hope it has nothing to do with the known syndrome 336 [that already runs in the family] because I would really find that terrible. I mean, 337 otherwise, it comes from me" – could you clarify?

Looking at table 1 there is a huge difference in when (after the diagnosis) the interview was conducted. I think it nice if the authors touched upon this in the results/discussion, as this may have affected the answers (families who are 3 years in will likely have come more to terms with the situation compared to families who are only 5 months in…).

Discussion:

Page 10. Could the authors elaborate on what benefits they experience having had by the family-wide approach? (this is not completely clear to me – as it is still ‘just’ parents who are interviewed – albeit not together – but not other family members? If I understand it correctly…)

Page 11. Under the reference to Afifi it states ‘competing desires for information’. This made me think that it may be valuable to add the regulations in you set up: does parents need to agree on the level of feed-back the want on the genetic analysis. Please add in the methods section.

Page 11. I think it could be valuable to add a reference/comment regarding disclosure to the extended family about the benefits of disclosing before results are in. One benefit could be: that you know what extended family members want to know? (a suggestion could be to find a relevant reference).

Page 13. I think it great that the authors come up with a recommendation regarding family centered counselling. This could to some benefit be listed (a)-€ in a table? And maybe just one sentence about this being an optimal approach but of course also both costly and time consuming.

Reviewer 2 Report

Comments and Suggestions for Authors

The first sentence states that 10% of children have a cancer predisposition

NEJM (2015) Zhang et al. the estimate is 10-15% have a cancer predisposition

Discussion states that this is the first to explore how parents experience family-related impact of testing.  There have been numerous manuscripts that have described the family impact

Reviewer 3 Report

Comments and Suggestions for Authors

I would like to thank the authors for the opportunity to review this work. First and foremost, I’d like to state that this is an interesting qualitative study with original findings that can add further evidence in the field of pediatric oncology. In fact, this study well addressed the family-level impacts of germline genetic testing for cancer predisposition and highlighted the key role of parents in their children’s clinical trajectory. Studies that include family factors and processes in this field are still scarce, so we need further knowledge to get a more comprehensive picture of the phenomenon, something this work can help with. Furthermore, the authors have made a successful effort to ground the data analysis on a widely recognized methodological framework (i.e., Multi Family Member Interview Analysis) that is appropriate to the population and phenomenon under study.  Congratulations! Despite that, there are some weaknesses and potential flaws that may compromise the quality of the submitted manuscript. Therefore, I would like to give the following suggestions and reflections, which I hope will make a valuable contribution to improving the quality of the manuscript.

GENERAL SUGGESTIONS

1.        I suggest that authors consult the following reference to ensure that all the essential steps in qualitative studies are properly described: O’Brien, B. C., Harris, I. B., Beckman, T. J., Reed, D. A., & Cook, D. A. (2014). Standards for Reporting Qualitative Research. Academic Medicine, 89(9), 1245–1251. https://doi.org/10.1097/acm.0000000000000388. It would be important to provide more details about the study design, setting, and researchers’ characteristics. For example, do the authors have previous experience in conducting qualitative research in this field and/or have training in qualitative methods? Why was this design and method chosen?

2.        The ‘Authors’ Contributions’, ‘Funding’ (this information is wrongly described in the ‘Acknowledgments’, so please replace it), ‘Institutional Review Board Statement’, ‘Data Availability Statement’, and ‘Conflicts of Interest’ sections are missing.

3.        Thank you for providing the interview guide in supplementary material, it is very useful information and increases transparency. However, I would suggest providing an additional translated version (in English).

ABSTRACT

The abstract is well-structured and concise. Nevertheless, some aspects should be reworded to make it clearer for readers:

4.        Line 26: When you write “family - as a whole -” I suggest you use the term “nuclear family”, as it is clearer.

5.        In the method, you should mention the study design employed and where the data was collected. For example, “This qualitative study was conducted in…”.

6.        Line 28: It would be important to add the mean age of the children followed by the standard deviation when you mention that “both parents were interviewed individually”.  For example, “(…) both parents of six ill children (five boys) with an average age of [complete the sentence] were interviewed individually”.

7.        Lines 28-30: Was the stepfather included in this process? It would be important to clarify this sentence and mention that not only biological parents took part in this study.

8.        In the results, you should mention the number of themes that emerged from the data analysis and address them briefly.

INTRODUCTION

9.        Lines 48-49: “at least 10% of children with cancer harbor a cancer predisposition syndrome (CPS)” Where? Worldwide? Please add this information. The same when you mention that germline genetic testing is progressively being integrated into standard pediatric oncology care (lines 52-53).

10.  Line 53: When you mention the medical benefits of germline genetic testing, you should elaborate a little more on the cost-effectiveness of this type of genetic testing. Furthermore, it would be relevant to describe what germline genetic testing refers to and mention, from an ethical point of view, what issues it raises.

11.  Lines 53-54: What kind of psychological impact does it have? Is it the same for patients and their families? Are there differences between mothers and fathers? Only the third paragraph answers some of these questions, so you should rephrase and reposition it to make it less confusing to read. With this information, the transition to the next paragraph, which focuses on the impact of childhood cancer on the child and the family, becomes less abrupt.

12.  Line 70: When you write “negative emotions”, you probably mean negative reactions.

13.  Lines 74-75: This sentence is unsourced. Please provide some references and examples.

14.  Lines 80-83: It is important to refer to the qualitative studies that exist on this topic, especially when the first paragraph of the discussion states that this study is “one of the first” that explores how parents experience the family-related impact of germline genetic testing. What evidence is available? Are there contradictory results? What are the specific gaps in the literature? This information will add quality and rigor to the state of the art. I recommend reading the following systematic review: Hunter, J. D., Hetherington, K., Courtney, E., Christensen, Y., Fuentes-Bolanos, N., Bhatia, K., & Peate, M. (2024). Parents’ and patients’ perspectives, experiences, and preferences for germline genetic or genomic testing of children with cancer: A systematic review. Genetics in Medicine, 26(9), 101197. https://doi.org/10.1016/j.gim.2024.101197

MATERIALS AND METHODS

15.  Lines 89-90: Were both parents interviewed on the same day? How long after? Under what circumstances (e.g., was there communication between the parent who was interviewed first and the other parent before the latter was interviewed?)? Please provide more details on these aspects.

Participants

16.  Is it possible to provide sociodemographic data on these parents other than their nationality? For example, age, level of education, ethnicity? These types of variables can influence parents’ responses and should be taken into account when interpreting the results. If you don't have this data, mention it in the limitations of the study.

17.  What was the reason for including a stepfather in this study (family 4)? Has the stepfather undergone this test? Please clarify these questions and explain how this can affect the results.

Data collection

18.  As mentioned above, It would be beneficial to provide more details about the setting/site (e.g., in what environment/place were the interviews conducted? What kind of services are available to these parents and children, such as psychological support? How is communication established between health professionals and families? Is there a family support coordinator?), as well as the researchers’ characteristics (e.g., previous experience conducting qualitative research within pediatric oncology, training in qualitative methods, gender). Furthermore, I was curious to know if any of the authors treat/intervene (e.g., psychological intervention) with patients or their families. If so, how did this affect the study? Please mention this in the limitations of the study.

19.  Line 118: “(…) was offered in parallel with the medical component of the study”. Please clarify what you mean by the “medical component of the study”. What does it include?

20.  Lines 133-137: The interview guide was developed by whom and based on what (e.g., literature review, other interview guidelines within pediatric oncology)? Please add this information.

Data analysis

21.  Lines 143-145: Did S.V.H. and S.H. both analyze 100% of the interviews? This information must be explicit. In addition, it is important to indicate the paradigm in which the authors who analyzed the data position themselves (e.g., positivism, post-positivism).

22.  Please indicate the Inter-rater agreement, preferably using the Kappa Statistic.

23.  One of the most worrying issues in this study is the fact that it is never mentioned, directly or indirectly, which criteria were used to ensure data saturation. This is critical in qualitative research. Please describe which criteria were followed and add the corresponding reference. 

RESULTS

24.  Figure 1: In the theme “Differences in coping with germline genetic testing between partners” you wrote “germinle”. Please correct the typo.

25.  When I read the results, I was a little confused regarding the differences between the theme ”Differences in coping with germline genetic testing between partners” and “feelings of guilt and mutual forgiveness”. Forgiveness is an emotion-focused coping strategy. I suggest to merge these two themes into one. For example, “Differences in coping and emotional reactions (in order to include feelings of guilt) between partners”.

26.  Why does theme 1 have an introduction before the description of the subthemes and themes 4 and 5 do not (lines 197-200)? You should maintain the same narrative logic in all cases.

DISCUSSION

27.  Line 435: “(…) and the potential long-term implications of germline genetic testing”. Please give some examples.

28.  Lines 436-441: Does previous research on this topic converge or diverge with these results? Is this an original finding? Please explain why you find this result interesting according to the available evidence.

29.  Lines 464-464: How? Are there any guidelines to follow?

30.  Line 489: “(…) the theme of “being together as a family matters (more)” What is this theme? Does it refer to theme 2?  

31.  Lines 526-528: Based on the results found, what kind of support can be offered? For example, would it be beneficial to promote dyadic coping? Adopt a family-centered empowerment model? What are the best practices within pediatric oncology care? Please elaborate.

CONCLUSIONS

32.  The authors must briefly mention the original findings of this study and its implications for practice and further research.

REFERENCES

33.  References should be revised according to the style required by the journal. For example, the year of publication should be in bold, not the volume, and the abbreviated journal name should be used.

Comments on the Quality of English Language

34.  Minor editing of English language required.

Round 2

Reviewer 3 Report

Comments and Suggestions for Authors

Dear authors,

The revisions made have notably enhanced the quality of the manuscript, especially in the materials and methods section. Furthermore, the authors' efforts to address the reviewers' comments in a thoughtful and reasoned manner are highly commendable.

I have no further comments and extend my congratulations on the successful completion of your article.

Yours sincerely,

The reviewer